# Reliable Characterization of Organic & Pharmaceutical Compounds with High Resolution Monochromated EEL Spectroscopy

**DOI:** 10.3390/polym12071434

**Published:** 2020-06-27

**Authors:** Partha Pratim Das, Giulio Guzzinati, Catalina Coll, Alejandro Gomez Perez, Stavros Nicolopoulos, Sonia Estrade, Francesca Peiro, Johan Verbeeck, Aikaterini A. Zompra, Athanassios S. Galanis

**Affiliations:** 1NanoMegas SPRL, Boulevard Edmond Machtens 79, B1080 Brussels, Belgium; alex@nanomegas.com (A.G.P.); tgalanis@nanomegas.com (A.S.G.); 2Electron Crystallography Solutions SL, Calle Orense 8, 28020 Madrid, Spain; 3EMAT, University of Antwerp, Groenenborgerlaan 171, 2020 Antwerp, Belgium; giulio.guzzinati@uantwerpen.be (G.G.); jo.verbeeck@uantwerpen.be (J.V.); 4LENS-MIND, Department of Electronics and Biomedical Engineering, Universitat de Barcelona, 08028 Barcelona, Spain; ccollbenejam@ub.edu (C.C.); sestrade@ub.edu (S.E.); francesca.peiro@ub.edu (F.P.); 5Institute of Nanoscience and Nanotechnology (IN2UB), Universitat de Barcelona, 08028 Barcelona, Spain; 6Department of Pharmacy, University of Patras, 26504 Patras, Greece; azompra@upatras.gr

**Keywords:** TEM, EELS, organic, pharmaceutical, plasmon loss, monochromator, amorphous

## Abstract

Organic and biological compounds (especially those related to the pharmaceutical industry) have always been of great interest for researchers due to their importance for the development of new drugs to diagnose, cure, treat or prevent disease. As many new API (active pharmaceutical ingredients) and their polymorphs are in nanocrystalline or in amorphous form blended with amorphous polymeric matrix (known as amorphous solid dispersion—ASD), their structural identification and characterization at nm scale with conventional X-Ray/Raman/IR techniques becomes difficult. During any API synthesis/production or in the formulated drug product, impurities must be identified and characterized. Electron energy loss spectroscopy (EELS) at high energy resolution by transmission electron microscope (TEM) is expected to be a promising technique to screen and identify the different (organic) compounds used in a typical pharmaceutical or biological system and to detect any impurities present, if any, during the synthesis or formulation process. In this work, we propose the use of monochromated TEM-EELS, to analyze selected peptides and organic compounds and their polymorphs. In order to validate EELS for fingerprinting (in low loss/optical region) and by further correlation with advanced DFT, simulations were utilized.

## 1. Introduction

Many pharmaceutical compounds may exist as polymorphs (e.g., they crystallize into different packing arrangements having the same chemical formula). Drug polymorphism is critically important in the pharmaceutical industry, as many of the solid-state properties of a compound are dependent on its polymorphic form. As an example, different polymorphic phases dissolve at different rates (solubility, bioavailability) affecting the adsorption of the compound in vivo, making it essential to control which polymorphic form is dosed to patients [1,2]. There is hence a requirement that the polymorphic behavior of a drug compound is thoroughly investigated and understood. 

Besides, when a drug or active pharmaceutical ingredient (API) is approved to be released into the market to be used for medication purposes, it is important that the marketed polymorphic form is well characterized (e.g., rules under US Food and Drug Administration—FDA in United States and European Medicines Agency—EMA in European countries). In practice, the approved API together with its other polymorphic forms is protected by Patent/IP rights before being launched onto the market and the released pharmaceutical product must contain exclusively the specific approved API form. Therefore, it is very important to characterize all the forms, not just only the marketed ones, as it might be possible that some polymorphic forms could have (or not) the same therapeutic effects as the marketed API, or it can be harmful for health and the environment [3].

On the other hand, during synthesis/production of a specific API in a particular polymorphic form, sudden transformation to another (more stable) polymorph may occur (e.g., the case of Ritonavir API where the marketed polymorph converted with time to a more stable but much less soluble form) [4,5]. Such transformed polymorphs may present in “trace” quantities that can also co-exist along with the main compound and are often unobserved during initial screening. As previously mentioned, the detection of impurities from the initial synthetic material or during the production process is also relevant. Therefore, there is a need to develop new reliable techniques and approaches to detect such trace polymorphic impurities (or other impurities) if present.

In the current “state of the art”, the standard characterization of an API (crystalline or amorphous), its polymorphs and its blend within a polymer matrix is usually performed using conventional or Synchrotron X-Ray powder diffraction (XRPD) (detection limit of 0.01 wt %). Other techniques like Fourier-transform infrared spectroscopy (FT-IR)/Raman scattering, differential scanning calorimetry (DSC) and solid-state nuclear magnetic resonance (NMR) can also be used with limited spatial resolution (from several hundreds of microns to nm); for example, Raman spectroscopy can only distinguish compounds at micron scale and photothermal-induced resonance (PTIR) has a spatial resolution reaching up to ~100 nm [6,7,8,9,10,11].

In the case of crystalline materials, electron diffraction (ED) in a TEM is ideal to identify whether an individual crystallite may belong to a new polymorph phase or not [12,13,14,15,16] at very local scale (10–20 nm) by collecting experimental diffraction patterns from an area of the sample and comparing them with the simulated diffraction patterns of well-known phases; electron diffraction based 3D tomography [17,18,19,20,21] can also be applied to determine the ab initio unit cell and crystal structure of each individual crystallite but its application can be very time consuming because of the significant number of API nanocrystals that may exist in a ng quantity powder sample; therefore there is a need to develop a technique to finely screen and detect (at nm scale) different APIs having various possible polymorphs (independent of crystalline or amorphous phases) and/or screen various possible phases in amorphous solid dispersion (ASD) [22], with nm spatial resolution. In the case of an amorphous material electron pair, distribution function can also be used to study pharmaceutical polymorphs [23].

It was recently shown that electron energy loss spectroscopy (EELS) in the low loss regime (0–50 eV) can be a suitable technique to distinguish amorphous organics at very local (nm range) scale [24]. It has been shown that EELS spectroscopy might be used to quantify local concentrations of API drugs through the amorphous polymer matrix with high accuracy at sub-100 nm resolution in a thin-film-like sample by recording the spectral signatures of the different compounds. In the work of Ricarte et al. [24], analysis of phenytoin/HPMCAS ASDs showed that drug and polymer were intimately mixed throughout the ASD, even at high drug loadings. 

Therefore TEM-EELS spectroscopy appears as a potentially complementary tool to identify and screen with high spatial resolution a number of organic small molecules (APIs, polymers in ASD) that otherwise cannot be distinguishable by other analytical techniques. In this work we try to explore further the possibility of finely characterizing various crystalline & amorphous APIs (including several polymorphs) by using high energy resolution monochromated STEM-EELS.

With EELS, we can study the loss of kinetic energy that electrons passing through or near a sample experience by exciting the sample itself. In particular in the low-loss regime the electron beam probes the optical transitions of the material in the ultraviolet to infrared range, providing information similar to that of optical UV−Vis spectroscopies. The spectra are approximately described by the so-called energy loss function Im[−1/ɛ~(ω)] (ELF).

## 2. Materials and Specimen Preparation/Instrumental Configuration—Data Collection

To identify APIs in crystals and ASDs we acquired spatially resolved maps of EELS spectra, through the so-called STEM-EELS spectrum imaging mode in the Titan3 microscope from Thermo Fisher Scientific installed at EMAT-Antwerp and equipped with a Wien filter monochromated electron source, a probe aberrations corrector and a Gatan Enfina electron spectrometer, featuring a fast electrostatic shutter. As organic compounds are very beam sensitive (critical dose for organics varies from 10–120 e/Å^2^) [25,26], to reduce beam damage, EELS low loss spectra from micron-sized grains were collected at 300 kV in low dose condition at RT with an effective resolution of 0.2 eV. The low-loss scattering events have a much higher cross section than higher energy transitions (e.g., the promotion of core-shell electrons to the valence levels) thus helping minimize the dose.

Using optimized low dose data acquisition, EELS data was acquired in STEM mode without any cryo-cooling techniques and no beam damage has been observed in our samples. We used a low convergence semi-angle (~0.5 mrad) to reduce current density within the probe, a collection semi-angle of 25 mrad. Since high energy electrons can induce radiation damage even at a distance of a few nm from its trajectory, we acquired the spectra from widely spaced (~50 nm) probe positions to avoid cross-damage between neighboring positions [27]. Each spectrum was acquired for 20 msec, where the scans covered areas between 0.5 and 2 μm^2^ and the dose was estimated to be 1 e/Å^2^ sec. Beam damage for the crystalline sample was monitored by observation of the high angle reflections in the diffraction pattern; the typical lifetimes of crystals were of the order of minutes in the current measuring conditions. Progressive amorphization by the electron beam proceeded via progressive loss of long-range order (fading of the high angle diffraction spots in crystalline APIs). For noncrystalline material the beam damage was monitored by observing any possible sample shape change in the STEM image. In addition, possible sample beam damage was also monitored by observing any low loss spectral change (1–5 eV EELS region) over time.

EELS data from several pristine sample areas were collected to compare the reproducibility of our results. EDX data were also collected from all samples to confirm the elements present in the molecule (especially for peptide TH_15 and TH_27: C, N, O, S) and to check for the presence of impurities (Figure 1). The EDX spectrometer for data acquisition is a Bruker Super-X, which uses 4 windowless detectors to reach a total collection angle of ~1 Steradian (Sr). The windowless design allows the effective detection of the signal of light elements, while the high acceptance angle greatly increases the dose-efficiency (detecting a higher fraction of the X-rays emitted by the sample), allowing it to perform EDX mapping even for organic materials. 

The EELS data were treated with the python package Hyperspy [28]. We applied, in order, a Savitzky–Golay filter, then a Richardson−Lucy deconvolution not with the purpose of increasing the energy resolution, but with that of reducing the impact of multiple scattering. We then removed the zero loss by fitting it to a pseudo-Voigt shape, subsequently subtracted from every point the spectral signature corresponding to the support film. From this pretreated data, we manually selected the thinnest area in order to obtain the best quality spectra.

Samples used for the experiment were the following: beta-cyclodextrin, hexacarboxy cyclohexane, tannin, peptide TH_15, peptide TH_27 and piroxicam form 1 and form 2 and aripiprazole form 2 and form 4 (Appendix A). All examined samples were crushed gently between two glass plates and then the powder sample was sprinkled on continuous carbon TEM grid. The size of the particles was relatively large (μm size) and in general samples were found thick in most areas; EELS observations were performed only in thin sample areas.

## 3. Results

For the synthetic peptide TH_15 and TH_27, fluorine impurity traces were detected by EDX (Figure 1); such traces might have come from the synthesis procedure and more specifically from the use of trifluoroacetic acid (CF_3_CO_2_H) during the cleavage of the peptide from the solid support (resin) [29].

In order to establish a fairly conclusive screening and identification methodology to readily differentiate among various organic molecular crystals and their polymorphs, we first analyzed by EELS various “reference” organic crystals; in most of the cases several characteristic low loss < 20 eV peaks were identified which were entirely specific to each organic compound (Figure 2).

From our obtained results it seemed that the use of TEM–EELS (Figure 2) with a monochromatic source (0.2 eV resolution) was necessary to reveal fine spectral details in the low loss EELS region for organic compounds, otherwise without its use many spectral fine details would not have been clearly distinguishable (for example the EELS spectra of tannin). All studied compounds (with the exception of the beta-cyclodextrin sample) revealed a different unique EELS signature at the low loss/optical region arising from π–π* transition peaks intimately tied to molecular structure. Based on our initial results it seemed that the EELS fingerprint (centered on the low loss/plasmon loss region) applied to organic compounds could potentially differentiate between them. These interesting results confirmed further early results obtained with low resolution EELS spectra on organic compounds [24].

In order to evaluate whether EELS could be used to distinguish between different polymorphic forms of the same compound, we acquired spectroscopic data from two different prepared forms of piroxicam and two different prepared forms of aripiprazole [30,31]. By choosing spectra from the thinnest parts of the analyzed samples (mean free path λ = 3, for both samples) for piroxicam, we were able to extract spectra showing a clear difference between the two forms of piroxicam (Figure 3). Though this mean free path (mfp) was on the limit for low loss EELS analysis (for qualitative mapping with edges < 800 eV the thickness should be between 0.1–1.2 mfp and for low-loss EELS analysis the thickness should be between 0.1–3.0 mfp), as the relative thickness was the same for both samples the results in this case were trustworthy and it was also reproducible [32]. The relative thickness was estimated using Gatan DM software using the log-ratio (relative) method [33]. This difference tended to disappear when thicker regions were analyzed suggesting that multiple scattering was a limiting factor for EELS to be sensitive enough to differentiate polymorphic samples. Such a clear difference, though, could not be observed in the case of the two polymorphic forms of aripiprazole samples even the EELS data were extracted from the thinnest part of the sample. During the EELS fingerprinting it was advisable to estimate the thickness in terms of mfps from the area where low loss EELS data were extracted; otherwise thickness variation might affect the result.

In order to have a deeper insight into the EELS low loss difference between various polymorphs of the same API compound, we studied theoretical simulations of the electronic structures of triclinic and monoclinic polymorphs of piroxicam API where there was a clear difference in their respective experimental EELS spectra (Figure 3).

Theoretical calculations of the electronic structure of the two piroxicam forms were performed using the plane wave plus local orbitals (APW + lo) method of DFT, implemented on WIEN2k code [34,35,36,37]. The exchange–correlation functional used was the Perdew–Burke–Ernzerhof method [38]. Crystallographic information of both phases was used as input. According to the bibliography, the piroxicam form 1 corresponded to a monoclinic phase with P21/c space group and 36 non-equivalent atomic positions. On the other site piroxicam form 2 corresponded to a triclinic phase (P1 space group) with 78 non-equivalent atomic positions [30]. The self-consistent cycle converged to 10^−5^ eV and the residual forces on atoms were below 0.01 eV/Å with 1000 k-points for both crystallographic phases. Complex dielectric function (CDF) and ELF were computed using OPTIC from WIEN2k.

The obtained energy bandgaps for both phases were underestimated with respect to the experimental values as is usual in DFT. The simulated data was calibrated using a scissor operator [39], the needed shift was computed using the first signal of the experimental data (common to both forms) placed at 3.825 eV.

In Figure 4, the real (dashed line) and imaginary (dotted line) part of the dielectric functions of both forms are plotted. Form 1 presented a plasmon resonance at 21.56 eV, ℜ(ε) = 0, and three main interband transitions, first peaks of the ℑ(ε); form 2 presented the plasmonic transition at a slightly higher energy, 20.53 eV and a set of three interband transitions could be detected. Comparing both forms, a clear difference in the relative intensity of the first transition was detected.

From the CDF the ELF can be directly obtained. Figure 5 shows the ELF compared with the experimental low loss spectra [40]. The difference in relative intensity on the imaginary CDF caused form 2 to have broader peaks for low losses of energy.

Looking in detail at the interband transitions (Figure 6), we concluded that the simulated data presented a good agreement with the experimental EELS data. The three main signals appeared on both, however peak 3 had too low intensity to be detected in any of the forms, peak 1 and 2 had the exact energy position for both phases and the peak 4 presented a redshift on form 2.

In view of our current results, we concluded that while it was still impossible to completely exclude the presence of residual thickness or damage effects in the experimental data, EELS could be used to detect the fingerprints of both polymorphs of piroxicam, through the change of the relative intensity of the first signal and the redshift.

## 4. Discussion

Detection and identification of various polymorphs/various different organic structures/APIs can be done with current Raman spectroscopy instrumentation with micron resolution; instead the “plasmon loss” EELS map technique achieves 10–50 nm spatial resolution and is far more general, as it may work for individual APIs and enable the efficient detection and screening of new pharmaceutical polymorphs during API synthesis, even in very low trace quantities (<0.1 wt %). Assuming that each organic compound/polymorph generally exhibits its unique (and distinguishable) π−π* transition plasmon loss peak, it should be possible to create “plasmon loss” maps (filtered at a particular loss peak, characteristic for each compound) that could potentially “differentiate” between various organic compounds/polymorphs. Such EELS maps are performed routinely for materials science applications (e.g., Si-Ge based layered semiconductors, core shell nanoparticles, perovskites etc.). Along with “known” basic/principal polymorph/crystal phase API, the generation of such “plasmon loss” EELS maps may reveal the existence of other possible polymorphic phases which may present <0.01 wt % in an ASD/formulated product where their further characterization could be done using electron diffraction tomography techniques [17,19] for crystalline materials or electron pair distribution function for amorphous materials [23].

Many pharmaceutical API formulations (up to 80% in their crystalline form) result insoluble in water and as a result their bioavailability is not adequate for appropriate medication; those API compounds in amorphous form show much higher bioavailability/solubility rates (up to 10–1600 times) [41] than their crystalline counterparts. As amorphous API forms, they cannot be conserved during long timescales (e.g., years) (which is mandatory for market acceptance) [42] in industrial practice poorly soluble crystalline APIs as prepared as a solid dispersion mixture between amorphous API and specific polymer mix (called ASD). This way the API/polymer mix ASD can be maintained in an “amorphous state” for longer time scales (e.g. years). Although the chemistry of API-polymer mix is not well-understood, it is generally assumed that in an “ideal” ASD, the API molecules are surrounded by polymer molecules without effective interaction between them. Again, in the case of ASD where both the API and polymer mix contain the same chemical elements (e.g. C, N, O) in their composition, low loss EELS mapping may enable detection of the “as early as possible” phase separation at ASD. It is also important to note that use of the scanned beam as in STEM and working in “low loss” part of the EELS spectra reduces the possible beam damage in beam sensitive (organic/pharmaceutical) compounds. The dose efficiency and sensitivity of EELS is also being further enhanced by using of direct detection cameras which are starting to gain acceptance in the TEM community. A low loss EELS spectrum is also useful to quantify water content in cells from various cellular components [43,44,45].

The challenge to overcome for generalized use of the EELS “low loss” technique is related to the fact that (in many cases) EELS low loss energy differentiation between various organics is often below 0.9 eV (conventional resolution of Gatan EELS spectrometers without a monochromated electron source). Therefore, the use of advanced TEM microscopes having EELS with a monochromated electron source (energy resolution < 0.2 eV, down to 0.01 eV in the best examples) seems mandatory to distinguish between various compounds.

Inelastic scattering between a fast electron and sample can occur even if the beam passes at a slight distance from the sample, a phenomenon known as inelastic delocalization. This phenomenon fundamentally induces a physical limit on the spatial resolution of the inelastic signal in EELS, even if the signal were recorded with an ideal instrument. Inelastic delocalization increases for higher energy beams and decreases with lower energy losses [32]. The inelastic delocalization factor R can be estimated by the root mean square impact parameter d_rms_ proposed by Pennycook [46] and it is ~9 nm for 3 eV loss with 300 kV beam and ~4 nm for 7 eV loss with100 kV beam. In most of the cases drug crystal nucleation or phase separation in ASD happens at 10–100 nm scale, so the inelastic delocalization in EELS will not be a hindrance to observe phase separation or drug nucleation in ASD [9,47,48].

While Cherenkov radiation has been a limiting factor in some EELS applications (e.g., bandgap measurements in semiconductors), it was not a limiting factor in low loss EELS characterization for organic compounds [49]. The refractive index of the studied compounds were not particularly high (e.g., ~1.6 β-cyodextrin, ~1.4 for hexacarboxy clyclohexane, ~1.9 for tannin, ~1.7 for piroxicam, ~1.6 for aripiprazole) meaning that while, depending on the beam energy, the electron velocity could be above the Cherenkov threshold, the emission was not going to be strong enough to significantly distort the fingerprints of the different organic compounds, and would at most give a small contribution to the featureless background.

## 5. Conclusions

The present work further confirms TEM-EELS as a potentially powerful complementary analytical tool to identify and screen with high spatial resolution organic small molecules (APIs, peptides, polymorphic forms of APIs) that otherwise cannot be distinguishable by other analytical techniques with that spatial resolution. Further work to be performed towards increasing sensitivity of that method to distinguish all possible polymorphic forms of an organic compound. It is also important to bear in mind that in order to get reliable EELS spectra for phase identification, sample thickness should be small enough (approx. <100 nm) to reduce multiple scattering effects. To obtain further experiment improvement to avoid such effects, appropriate sample preparation protocols have to be developed, e.g., with ultramicrotomy or Cryo FIB.

## Figures and Tables

**Figure 1 polymers-12-01434-f001:**
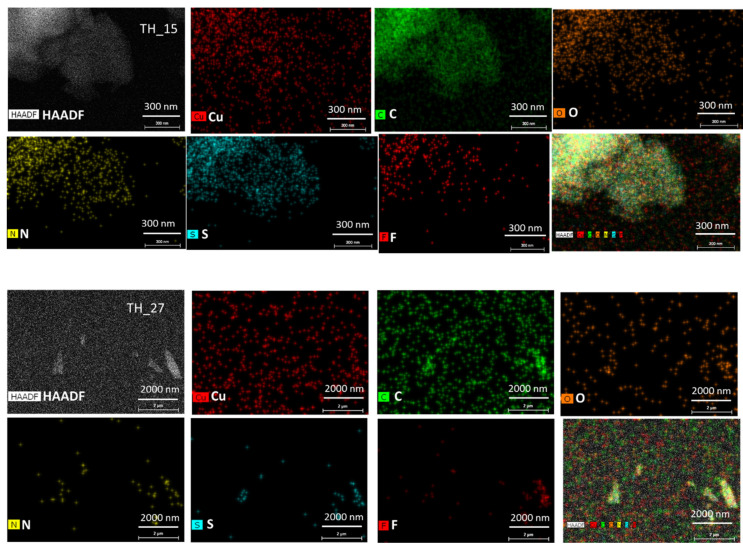
Scanned area and EDS mapping of TH-15 (**up**) and TH-27 (**down**) peptide compound shows presence of presence of C, N, S, O (from the molecule) and F (impurities). Cu signal is homogeneously distributed and is an artifact due the TEM sample grid being made of copper. Top left corner image shows the high annular dark field image (HAADF) of the observed area.

**Figure 2 polymers-12-01434-f002:**
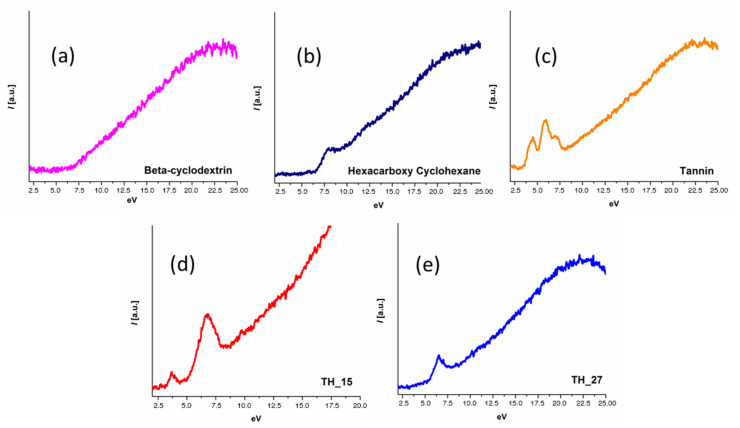
Low loss electron energy loss spectroscopy (EELS) for various organic small molecules showing characteristic (π–π* transition peaks) in low loss region, for (**a**) beta-cyclodextrin with no characteristic signal; (**b**) hexacarboxy cyclohexane ~8 eV; (**c**) tannin ~4.5, ~4.9, ~7.2 eV; (**d**) peptide TH_15~3.8, ~6.9 eV, (**e**) peptide TH_27~6.5 eV peak was observed.

**Figure 3 polymers-12-01434-f003:**
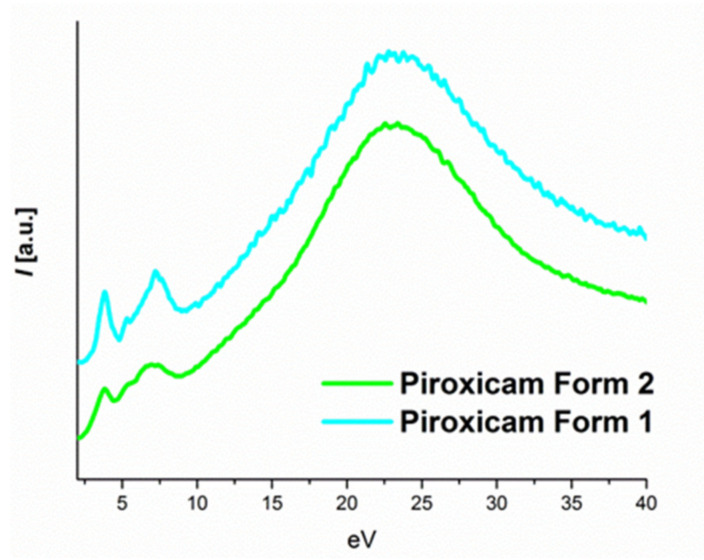
High resolution low loss EELS spectra from piroxicam form 1 (monoclinic crystal structure) and form 2 (triclinic crystal structure).

**Figure 4 polymers-12-01434-f004:**
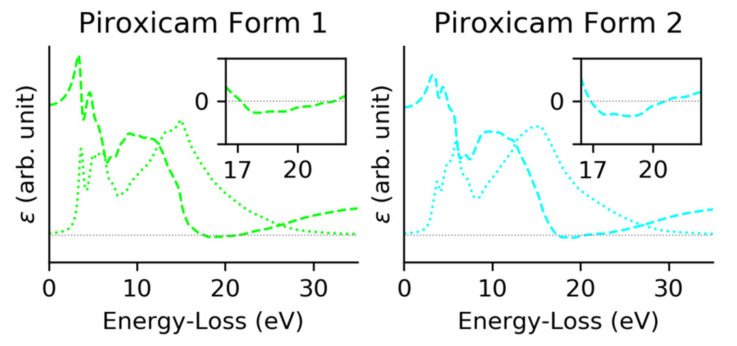
Complex dielectric function of monoclinic form 1 (**left**) and triclinic form 2 (**right**) phases of piroxicam. The dotted line corresponded to the imaginary part of the CDF and the dashed line to the real part. The insets corresponded to a zoom of the zero-crossing energies, ℜ(ε) = 0.

**Figure 5 polymers-12-01434-f005:**
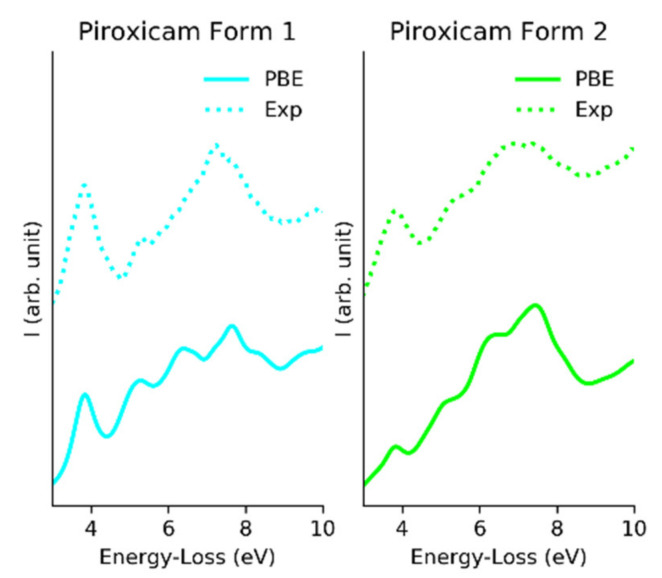
Comparison between calculated EELS spectra (solid line) and experimental EELS data (dotted line), for phases form 1 and form 2.

**Figure 6 polymers-12-01434-f006:**
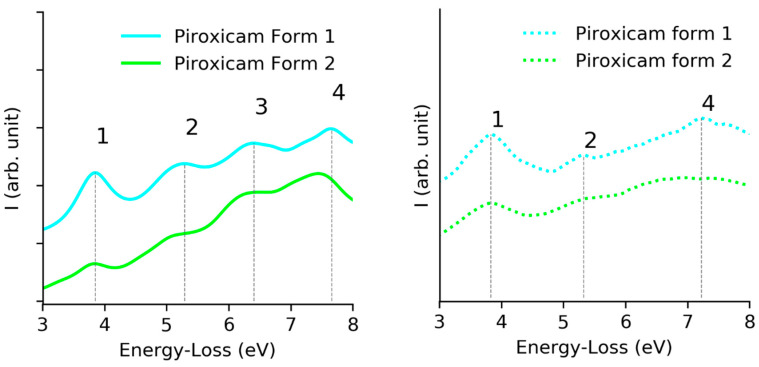
Comparison of the interband transitions observed on theoretical (**left**) and experimental data (**right**) for both forms of piroxicam.

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
