# Peer review of "Reliable Characterization of Organic & Pharmaceutical Compounds with High Resolution Monochromated EEL Spectroscopy"

_polymers, 2020, doi:10.3390/polym12071434_

Round 1

Reviewer 1 Report

The authors demonstrated how effectively EELS fingerprinting technique, in particular with low-loss regions, can be applied for the analysis of selected peptides and their polymorphs. I think this technique would be very powerful and attractive when amorphous materials are studied due to lack of characteristic diffraction patterns resulting from short range order of atoms in amorphous materials.

1. From the title, I suggest that authors use proper acronym for EELS. In the "EELS" the spectroscopy is included already. Therefore, "EEL spectroscopy" is correct expression. Please change it.

2. Line 28-29, the sentence should be divided into two sentences:
and their polymorphs. In order to validate EELS for fingerprinting and by further correlation with advanced DFT simulation were utilized (or used).

3. Line 43, Therefore it is -> Therefore, it is

4. Line 80, the reference for Ricarte et al. is missing

5. Line 990, Titan 3 is the local custom name for Titan microscope from Thermo Fisher Scientific at EMAT. Please use the full name from the company correctly.

6. From line 116-123, I am not sure why authors used EDX technique to investigate C, N, S, O in the specimen, because EELS technique is much more beneficial for such light elements.

7. Authors selected one spectrum from the thinnest area (?) to show good agreements with theoretical calculations without showing the measured relative thickness values (t/lambda) where they spectra acquired. In addition, from the line 168 -171, they wrote the difference tends to disappear when thicker regions are analyzed.... Then, I am not able to apply this technique because we don't know when the different fine structures disappear. Please, provide more information for other scientists.

8. line 194, there is no plasmon resonance at 20.53 in the Figure 4.

9. From line 208 -211, the authors should be careful with the interpretation from such fine structures without knowing the thickness sample, because the broadening due to thickness variation can lead to the change in the intensity of spectrum.

10. Line 250, the spreading of direct detection cameras -> using direct detection cameras

11. Line 256, Gatan EELS system has no monochromator, I think the authors would say that the monochromated beam(Wien filter) for the gun is used in the microscope.

12. line 265, to avoid dynamical effects -> to reduce multiple scattering effects.

In general, the manuscript is well prepared, but the authors should answer the above mentioned aspects with proper corrections before it is published.

Reviewer 2 Report

The authors provide a detailed account of their investigations into using Electron Energy Loss Spectroscopy (EELS) as a tool for identifying and differentiation different polymorphs of selected peptides and organic compounds. The EELS methodology use builds on established techniques in the materials science/nanotechnology community. While there are reports of similar work (see https://doi.org/10.1039/C7PY01459G), the current manuscript is an interesting proof of concept linking experimental work with the theoretically derived energy loss function (ELF). I find that the work performed is sound and I believe it will be of interest to the community, especially for work on amorphous phases. The manuscript is well written and to the point. I have three comments/suggestions that should be taken into account:

* I am missing a discussion of retardation effects/Cherenkov radiation in these materials. What is their refractive index, and should we expect such effects to contribute to the EELS spectra?

* The authors could comment more explicitly on the expected spatial resolution of the low loss EELS measurements, taking the inelastic delocalization into account, compared to the relevant length scales for these materials/samples.

* Figure 1 could be reworked to improve the legibility of the labels and scale bars.
